# The Oxidative Pathway to Dopamine–Protein Conjugates and Their Pro-Oxidant Activities: Implications for the Neurodegeneration of Parkinson’s Disease

**DOI:** 10.3390/ijms20102575

**Published:** 2019-05-25

**Authors:** Kazumasa Wakamatsu, Kenta Nakao, Hitomi Tanaka, Yuki Kitahori, Yui Tanaka, Makoto Ojika, Shosuke Ito

**Affiliations:** 1Department of Chemistry, Fujita Health University School of Medical Sciences, 1-98 Dengakugakubo, Kutsukake-cho, Toyoake, Aichi 470-1192, Japan; isis101515@yahoo.co.jp (K.N.); hitanaka@u-gifu-ms.ac.jp (H.T.); yuki728.apple73@gmail.com (Y.K.); yuit8414@gmail.com (Y.T.); sito@fujita-hu.ac.jp (S.I.); 2Department of Applied Biosciences, Graduate School of Bioagricultural Sciences, Nagoya University, Chikusa-ku, Nagoya 464-8601, Japan; ojika@agr.nagoya-u.ac.jp

**Keywords:** neuromelanin, pro-oxidant activity, bovine serum albumin, β-lactoglobulin, glutathione, hydrogen peroxide

## Abstract

Neuromelanin (NM) is a dark brown pigment found in dopaminergic neurons of the substantia nigra (SN) and in norepinephrinergic neurons of the locus coeruleus (LC). Although NM is thought to be involved in the etiology of Parkinson’s disease (PD) because its content decreases in neurodegenerative diseases such as PD, details are still unknown. In this study, we characterized the biosynthetic pathway of the oxidation of dopamine (DA) by tyrosinase in the presence of thiol peptides and proteins using spectroscopic and high-performance liquid chromatography (HPLC) methods and we assessed the binding of DA via cysteine residues in proteins by oxidation catalyzed by redox-active metal ions. To examine whether the protein-bound DA conjugates exhibit pro-oxidant activities, we measured the depletion of glutathione (GSH) with the concomitant production of hydrogen peroxide. The results suggest that the fate of protein-bound DA conjugates depends on the structural features of the proteins and that DA-protein conjugates produced in the brain possess pro-oxidant activities, which may cause neurodegeneration due to the generation of reactive oxygen species (ROS) and the depletion of antioxidants.

## 1. Introduction

Neuromelanin (NM) is a brown to black-brown insoluble melanin-like pigment present in the midbrain and is known to exist in humans, monkeys, dogs, horses, rats, frogs and other animal species [1,2]. NM is localized in dopaminergic neurons of the substantia nigra (SN) and in norepinephrinergic neurons of the locus coeruleus (LC) [3,4,5]. Melanin pigments in the skin, hair and eyes are derived from DOPA, whereas NM in the SN is derived from l-dopamine (DA) and NM in the LC is derived from l-norepinephrine (NE) [5]. Recently, a NM-like pigment was also found in other regions of the brain, including the putamen, premotor cortex and cerebellum [4,6]. However, in contrast to melanin in the skin, the structure and function(s) of NM in the brain remains unclear due to the difficulty of their isolation and the absence of suitable biochemical model compounds. NM is not excreted under physiological conditions and it is thought that there is no mechanism to completely decompose NM in situ [6]. NM starts to accumulate in the brain at ages 3 to 5 and continues to increase with age, but is markedly decreased in patients with Parkinson’s disease (PD) [3,7,8]. Thus, in recent years, NM has been implicated to play an important role in the pathogenesis of PD, but the details are still unknown. The role(s) of NM in the brain are uncertain, but their neuroprotective and neurotoxic effects have been proposed [6]. NM protects nerve cells from toxicity by scavenging metals (iron(III), copper(II), zinc(II), aluminum(III), lead(II) ions, etc.), toxic organic compounds and free radicals, [4,6,9]. NM also provides a protective mechanism against the toxicity of DA, preventing the cytosolic accumulation of cytotoxic dopamine quinone (DAQ) produced by DA oxidation [10]. Cytotoxic DAQ reacts with l-cysteine (Cys) to form Cys-DA in the process of NM production, resulting in its detoxification [10]. As a result, the formation of NM is thought to be the result of a cellular defense mechanism against high oxidative stress. On the other hand, from the aspect of neurotoxicity, NM becomes a source of compounds exhibiting free radicals and exerts its cytotoxicity through the degradation of pigmented neurons, perpetuating the cycle of oxidative stress and nerve cell loss [6,8,11,12]. Thus, the physiological function of NM is thought to be a complex two-edged sword.

Sulzer et al. demonstrated that NM is generated by the oxidation of cytosolic DA in cultures of dopaminergic neurons [10]. The production of NM is a complex and continuous process, leading to the accumulation of organelles containing NM in SN neurons. NM is formed from the oxidation of DA, followed by interactions with other cellular components such as small thiols (Cys and glutathione (GSH)), proteins, lipids, and metals, etc. NM is confined in cytoplasmic cell organelles and is covered with a bilayer membrane [10]. In addition, an X-ray crystal diffraction study revealed that isolated NM has a structure stacked on a Z plane having a stacked space like 4.7 Å graphite, which has quite different properties from melanins found in the skin, hair and eyes etc. [13,14]. In addition, this structure is very similar to that seen in the β-cross sheet structure of amyloid protein aggregates and has been shown to be a complex polymeric pigment with a multilayered 3D structure [14]. Regarding the structure of NM, it has been speculated that melanin pigments conjugated oxidatively from DA and Cys-DA are associated in complexes with aliphatic and peptidic compounds based on the results of structural studies of NM by several groups [4,15,16,17,18]. Other components include α-synuclein as a lipoprotein covalently bound to NM in the brains of patients with PD [15]. Although this process has a competitive pathway of the oxidative polymerization of DA to eumelanin, the intermediate quinone oligomers produced in this process can react quickly with proteins. It is known that, in the presence of Cys, the reaction with DAQ to form Cys-DA is faster than the intramolecular cyclization of DAQ. Oxidized DA binds to proteins through Cys residues, but it is still unknown how conjugates would be oxidized subsequently. Recent studies have demonstrated that the oxidation of DA produces oxidative products, leading to mitochondrial dysfunction, impaired protein degradation, α-synuclein aggregation into neurotoxic oligomers, and oxidative stress in vitro, and the DA content is critical for both DJ-1 knockout and A53T α-synuclein transgenic mice to develop PD pathological features, providing evidence for DA action in PD pathogenesis in vivo [19].

It is controversial whether tyrosinase is present in the brain. It is generally accepted that NM is formed by non-enzymatic DA autoxidation [20], in contrast to the well-known enzymatic synthesis of peripheral melanins [21]. Some studies did not detect the tyrosinase expression in the human brain [22,23]. Recently, conspicuous NM production was obtained solely by overexpressing human tyrosinase in the rodent brain [24]. This shows that further studies are required to determine in a definite manner a potential contribution of tyrosinase or other enzymes in NM synthesis. It has been suggested that the initial step of NM formation depends on the presence of fibrillar protein seeds, with which catecholamine quinones apparently react more readily than with other protein materials [25]. Recently, Ferrari et al. used synthetic analogues of NM obtained by the melanization of fibrillated β-lactoglobulin (βLG) containing 1 Cys residue and 2 cystine residues in order to develop a model for PD [26]. βLG is a 18.4 kDa protein containing a central β-barrel motif (the so-called calyx), which provides the ligand binding site, surrounded by 3-helices [27]. βLG contains an accessible and reactive Cys (Cys121) and can be converted into an amyloid fibrillar form (fLG) [28,29].

In this study, we prepared a water-soluble DA–protein conjugate by oxidizing DA with tyrosinase in the presence of bovine serum albumin (BSA), which contains 1 Cys residue and 17 cystine residues. We also examined the oxidation of DA in the presence of βLG. It is important to note that the DA melanization mechanism with BSA or βLG supports the emerging evidence for the initial steps of the biosynthesis of NM, where DA and its metabolites randomly attack the surface of a core of aggregated proteins in the cytosol. The oxidation of DA was also catalyzed by redox metal ions such as Fe(III) or Cu(II). Whether these compounds are covalently bound to DA via Cys residues in proteins was examined by the production of Cys-DA isomers by the HCl hydrolysis of DA-conjugated proteins (Figure 1). As it was unknown whether protein-bound DA conjugates exhibit pro-oxidant activities, we examined whether they can oxidize GSH, the most important cellular antioxidant, to oxidized glutathione (GSSG) with the concomitant production of hydrogen peroxide. This study shows that protein-bound DA conjugates formed during the production of NM have pro-oxidant activities that generate reactive oxygen species (ROS) and oxidize GSH to GSSG. We also examined in detail the melanin biosynthesis pathway by the oxidation of DA with various peptides and proteins using spectroscopic and high-performance liquid chromatography (HPLC) methods. The results demonstrate that the DAQ produced is bound to proteins via their SH groups and that the binding efficiency and the fate of DAQ depends on the different structural features of proteins.

## 2. Results

### 2.1. Spectroscopic Examination of the Tyrosinase-Catalyzed Conjugation of 4-Methylcatechol (MeCA) with Non-Protein and Protein Thiols and Related Compounds

We initially examined the reactivity of 4-methyl-1,2-benzoquinone (MeBQ) to avoid complexity due to the intramolecular cyclization of the DAQ side chain. Li et al. [30] have performed a detailed kinetic study on the initial phase of reactions of MeBQ with various proteins, thiols, and amino acids. In our study, we wished to follow the overall process of adducts formation and their decay during 60-min reactions. As shown in Figure 2A, the tyrosinase-catalyzed oxidation of MeCA at pH 7.4 produced MeBQ, which has an absorption maximum at 400 nm, typical for *ortho*-quinones [31]. MeBQ gradually decayed due to its conversion to a quinone methide [31]. In the presence of 2 mol eq. of non-protein thiols (i.e., Cys, GSH, *N*-acetyl-l-cysteine (NAC) and DPRA(Cys) (a heptapeptide containing 1 Cys residue, Ac–RFAACAA–COOH), stable conjugates having absorbance peaks at 290 nm were produced (data not shown). Those conjugates were produced by the addition of a thiol group to MeBQ, a reaction similar to that reported for cysteinyldopa isomers [32]. However, when the reaction was carried out in the presence of only 1 mol eq. of those thiols, the conjugates initially formed were oxidized to *ortho*-quinones, having absorption maxima at 520–540 nm. Among them, the GSH conjugate was almost stable during a 60-min reaction time with an absorption maximum at 544 nm, which may correspond to a stable *S*-conjugated *ortho*-quinone (Figure 2B). We then examined whether the amino acids l-histidine (His) and l-lysine (Lys) are able to react with MeBQ. His contains a reactive imidazole group while Lys contains an extra terminal amino group [30,33,34]. As shown in Figure 2C, 10 eq. *N*α-acetyl-l-lysine (NAcLys) reacted with MeBQ to produce a stable *ortho*-quinone conjugate with an absorption peak at 500 nm in 30 min as well as Lys (data not shown). Although His slowly reacted with MeBQ to produce a stable *ortho*-quinone conjugate with an absorption peak at 500 nm (data not shown), 10 eq. *N*-acetyl-l-histidine (NAcHis) or 10 eq. *N*-acetyl-l-leucine (NAcLeu) did not react with MeBQ (Figure 2D). The proteins BSA and βLG, each having 1 Cys residue, also reacted with MeBQ. As shown in Figure 2E, the reaction with BSA (2 eq. containing 0.6 eq. of Cys residue, see Materials and Methods) proceeded within 1 min to produce a stable *ortho*-quinone conjugate with an absorption maximum at 500 nm. The reaction with 3 eq. BSA produced only a trace (0.016 A) of this absorption peak, indicating that most of the conjugate remained in a reduced form. βLG (2 eq. containing 1.2 eq. of Cys residue) rapidly produced an *ortho*-quinone conjugate with an absorption peak at 492 nm, which then gradually decayed (Figure 2F). Whether those BSA and βLG conjugates were formed through their Cys residue or their amine residues (the imidazole group in His and/or the terminal amino group in Lys) will be examined later.

### 2.2. Spectroscopic Examination of Tyrosinase-Catalyzed Conjugation of DA with Non-Protein and Protein Thiols and Related Compounds

We next examined the conjugation of DAQ with thiols and related compounds. The tyrosinase-catalyzed oxidation of DA produced DAchrome (DAC) within 1 min (absorption peaks at 298 and 474 nm), which gradually decayed to a melanochrome-type compound with broad absorption maxima around 300 and 510 nm (Figure 3A). When the oxidation was carried out in the presence of 2 eq. of non-protein thiols, stable thiol conjugates having absorption maxima at 292 nm were produced (data not shown). However, in the presence of 1 mol eq. of those thiols, the oxidation proceeded through a complex series of reactions. As shown in Figure 3B, the reaction with GSH gradually produced a melanochrome having an absorption maximum at 370 nm. The reaction with NAC produced a melanochrome having an absorption maximum at 568 nm in 30 min (Figure 3C). Interestingly, the reaction produced a transient absorption maximum at 488 nm during the first 3 min, suggesting the production of an *S*-conjugated *ortho*-quinone intermediate. The reaction with DPRA(Cys) proceeded similarly as NAC, giving an absorption maximum at 492 nm in 1 min, which shifted to a peak at 582 nm in 30 min (Figure 3D). The reaction of DAQ with His, Lys (2 eq.) and NAcLys (10 eq.) was also performed to examine whether amino groups in those basic amino acids are able to compete with the cyclization (data not shown). The results show that spectra from the oxidation of DA almost superimposed with those in the presence of His, Lys (2 eq.) and NAcLys (10 eq.), which clearly indicates that the cyclization proceeds much faster than the addition reactions of amino groups. Likewise, DPRA(Lys), a heptapeptide without a Cys residue (Ac–RFAAKAA–COOH), did not react with DAC (data not shown). As shown in Figure 3E, the reaction of DA in the presence of BSA (2 eq.) produced DAC (absorption peak at 470 nm), although in a yield lower than in the absence of BSA (Figure 3A). The absorption maximum at 370 nm then gradually increased. The reaction with βLG (2 eq.) also produced DAC in a yield higher than with BSA (Figure 3F). The yield of DAC was compared at 1 min, giving 30% and 80% yields of DAC in the presence of BSA and βLG, respectively. When the reaction with BSA was carried out with 4 eq. BSA, the yield of DAC dropped to 15% and the absorption peak at 370 nm increased at a much slower rate than with 2 eq. BSA (data not shown).

It became clear that the reaction of DAQ with thiol groups in proteins is competitive with the cyclization of the side chain amino group giving DAC. Therefore, the reactivity of DAC against non-protein and protein thiols were examined. NAC (1 eq.), used as a representative non-protein thiol, reacted gradually with DAC in 10 min, producing a melanochrome chromophore with absorption peaks at 296 and 594 nm in 30 min (Figure 4A). The insert in Figure 4A shows that this reaction proceeds in two steps: the absorption peak at 480 nm decreases in 10 min, and then the absorption peak at 600 nm starts to increase, indicating that the production of 5,6-dihydroxyindole (DHI)-NAC conjugate is followed by the development of melanochrome chromophore. BSA (2 eq.) reacted gradually with DAC to produce melanochrome absorption peaks around 370 and 560 nm (Figure 4B). The absorption spectrum at 60 min was similar to that starting from DAQ and BSA (Figure 3E). However, the absorbance peak at 560 nm increased by 1.5-fold, providing additional support for the reactivity of BSA against DAC. By contrast, the reaction of βLG with DAC proceeded very similarly as DAC alone (Figure 4C). However, direct comparison of absorption spectra at 60 min shows some difference between the spectra of DAC in the presence or absence of βLG, with increases of absorption peaks around 370 nm and above 550 nm in the presence of βLG (Figure 4C). This suggests that at least some portion of DAC reacted with βLG to produce a melanochrome-protein. Therefore, we constructed a differential spectrum for the product of βLG with DAC, by assuming that 80% of the DAC was converted to DHI-melanin while 20% reacted with βLG. Also, some (assuming 30%) DAC may escape from reaction with BSA. The origin of the melanochrome absorption peak was then confirmed in the reaction of DHI with NAC (1 eq.). As shown in Figure 4D, an absorption peak at 302 nm due to the DHI-NAC conjugate appeared in 1 min (confirmed in the reaction of DHI in the presence of 2 eq. NAC), and absorption peaks around 314 and 590 nm gradually increased. Figure 4E compares the differential spectra for DAC + BSA (70% reaction) and DAC + βLG (20% reaction) at 60 min. The differential spectra had absorption maxima around 370 and 570–590 nm. The reactivity of DAC against His or Lys (2 eq.) was also examined. The results, confirmed that there was little reactivity, if any, of those amino acids with DAC (data not shown).

### 2.3. High-Performance Liquid Chromatography (HPLC) Analysis of Tyrosinase-Catalyzed Conjugation of 4-Methylcatechol (MeCA) and l-Dopamine (DA) with Bovine Serum Albumin (BSA) and β-Lactoglobulin (βLG)

As stated above, BSA and βLG react with DAQ to form conjugates in yields of 70% and 20%, respectively. Those conjugates are likely to be *S*-conjugates produced through Cys residues, because the amino acids His and Lys are not able to react with DAQ or DAC (data not shown). We next examined to what extent those conjugates liberate Cys-MeCA and Cys-DA isomers after HCl hydrolysis of the conjugated proteins. After reduction of the oxidation mixtures with NaBH_4_ to reduce *ortho*-quinone to the catechol moiety, proteins were hydrolyzed with 6 M HCl in the presence of phenol and thioglycolic acid [35]. The hydrolysates were then treated with alumina to extract catechols and were analyzed by HPLC with electrochemical detection.

At first, BSA and βLG modified with MeBQ were examined. As shown in Figure 5A, the reaction with BSA (2 eq.) at 3 min yielded 3-*S*-Cys-5-MeCA (corresponding to 5-*S*-CysDA in structure), 4-*S*-Cys-5-MeCA (corresponding to 6-*S*-Cys-DA) and diCys-MeCA in a combined yield of 66%. The total yield was decreased to 54% at 60 min. Interestingly, the ratio of 3-*S*-Cys-5-MeCA, 4-*S*-Cys-5-MeCA and diCys-MeCA was 98:0:2 in the hydrolysate of the free forms of Cys-MeCA isomers, while it was 45:41:13 from the BSA-MeCA conjugates. This indicates that the relative reactivity of the 4-position vs. the 3-position of MeBQ increased in the conjugation with proteins. The reaction of MeBQ with βLG (2 eq.) produced Cys-MeCA isomers in a combined yield of 16% in 3 min and of 8% in 60 min. The low yield of Cys-MeCA from βLG suggests that a major reaction proceeds through conjugation with His and Lys residues. In fact, βLG contains 2 His and 15 Lys residues while it contains only 1 Cys residue.

We next examined the modification of BSA (2 eq.) with DAQ. As shown in Figure 5B, the reaction at 3 min yielded 5-*S*-Cys-DA, 2-*S*-Cys-DA, 6-*S*-Cys-DA and diCys-DA in a combined yield of 46%. The yield did not change at 60 min (47%). Interestingly, the ratio of 5-*S*-Cys-DA, 2-*S*-Cys-DA, 6-*S*-Cys-DA and diCys-DA was 83:16:1:0 in the hydrolysate of the free forms of Cys-DA isomers, while it was 51:30:16:3 from the BSA-DA conjugates. As noted for the BSA-MeCA conjugates, the relative reactivity of the 6-position of DAQ vs. the 5-position increased. The reaction of DAQ with βLG (2 eq.) produced Cys-DA isomers in a combined yield of only 2.1% in 3 min and the yield did not increase in 60 min.

### 2.4. Time Course of Conjugation of DA with BSA Catalyzed by Fe(III) and Cu(II)

We then examined how BSA is modified when DA is oxidized by redox metal ions, Fe(III) and Cu(II). Ethylenediaminetetraacetic acid (EDTA) was added to promote the oxidation [36]. In the Fe(III)-EDTA-catalyzed oxidation of DA in the presence of BSA (2 eq.) (Figure 6A), the combined yields of Cys-DA isomers increased slowly and almost linearly and reached 29% after 4 h. The remaining DA was 67% after 4 h, indicating that the binding through the Cys residue proceeds almost quantitatively. In the Cu(II)-EDTA-catalyzed oxidation of DA with BSA (2 eq.), the reaction proceeded much slower compared to the oxidation with Fe(III) and the combined yield of Cys-DA isomers reached only 10% in 4 h (Figure 6B). When EDTA was omitted from the Fe(III)-catalyzed oxidation of DA in the presence of BSA (2 eq.) (Figure 6C), the combined yield of Cys-DA isomers dropped to 15% in 4 h. It appears that in the absence of EDTA, Fe(III) forms a stable complex with DA and makes the oxidation slower.

### 2.5. Pro-Oxidant Activities of Various Melanins Derived Oxidatively from DA-Thiol Conjugates

GSH (1 mM) was exposed to DA-thiol conjugates prepared by tyrosinase oxidation of DA (1 mM) in the presence of Cys (1 mM) or GSH (1 mM), after which the amounts of remaining GSH and oxidized glutathione (GSSG) were analyzed by our specific HPLC method after 30 min or 60 min [37]. We then examined whether H_2_O_2_ is produced during the oxidation of GSH by those synthetic melanins. H_2_O_2_ was measured by absorbance at 568 nm using Ampliflu™ Red reagent. After the preparation of melanin produced by tyrosinase oxidation of DA (0.5 mM) in the presence of BSA (0.5 mM) or βLG (0.5 mM), the pro-oxidant activity of protein-bound DA conjugates was evaluated by adding GSH (0.5 mM) as well. Figure 7A shows the amount of H_2_O_2_ after the addition of GSH to melanins prepared in situ. The amount of H_2_O_2_ gradually increased with time except for Cys-DA conjugate (Cys-DA-melanin). As a control, we measured the amount of H_2_O_2_ generated by DA-melanin and oxidative products from BSA and βLG alone. GSH-DA conjugated melanin produced higher amounts of H_2_O_2_ compared to other DA-conjugated proteins. Figure 7B shows the amount of GSH remaining and GSSG produced compared with the original amount of GSH (100%). DA-melanin showed the lowest activity. The GSH-DA conjugate showed the highest pro-oxidant activity with ca. 70% conversion of GSH to GSSG in 30 min, followed by BSA-*S*-DA (ca. 40% conversion) and βLG-*S*-DA (ca. 40% conversion). Cys-DA conjugate (Cys-DA-melanin) showed a lower activity (24% conversion). The mixtures produced by the tyrosinase oxidation of BSA and βLG generated 14% and 6% of GSSG after 30 min, respectively. Figure 7C shows the GSSG to GSH ratio as an oxidative stress marker. The GSSG to GSH ratio is higher in order of GSH-DA, BSA-*S*-DA, βLG-*S*-DA, and Cys-DA conjugates.

## 3. Discussion

Zucca et al. [6] showed that in the cytoplasm of SN neurons, DA is oxidized to quinone and semiquinone by catalytic iron ions, and those highly reactive compounds then react with assembled proteins that have accumulated in the cytoplasm. The polymerization begins to form a melanin-protein complex and also binds with high concentrations of metals, in particular Fe(III) ions. As a result, polymers that are not decomposed are self-phagocytized into vacuoles and taken up by lysosomes. After dissolution in lysosomes, they interact with lipids and other proteins with age to form organelles containing NM. 1,2-benzoquinones, formed by the oxidation of catechols such as DA, are highly reactive electrophiles. Catechol (1,2-dihydroxybenzene) structures are found in many naturally occurring polyphenols, and the formation of 1,2-benzoquinones in biological systems can thus be expected to accompany the antioxidative action of phenolic compounds. 1,2-benzoquinones undergo Michael addition reactions with nucleophilic groups such as thiols and amines in amino acids, peptides and proteins. Li et al. [30] examined the role and quantitative importance of reactions of MeBQ with thiols versus amines, guanidine nucleophiles present in amino acids, peptides and proteins. MeBQ reacted rapidly with BSA and the second order rate constant was at least 12-fold greater than α-lactalbumin, which does not contain any Cys residues. A similar result was also obtained in our experiments using DPRA(Lys) (data not shown). The amino acids His and Lys were not able to react with DAQ or DAC (data not shown). Li et al. showed that low-molecular mass thiols (Cys, NAC and GSH) react much faster (~68-fold or higher) with MeBQ than do thiol containing proteins (such as BSA) [30]. Those results demonstrate that MeBQ has a much higher reactivity toward SH residues than NH_2_ residues.

Zhou et al. [38] demonstrated that when GSH reacted with DAQ, short-lived intermediate GSH conjugates (5-*S*-GS-DAQ and 2-*S*-GS-DAQ) formed initially. Those intermediates cyclize spontaneously to form reactive 7-*S*-GS-DAC although 5-*S*-GS-DA and 2-*S*-GS-DA are formed under sufficiently reductive conditions. 7-*S*-GS-DAC, which has an absorption peak at 375 nm, is so reactive that it can further conjugate with another GSH to form 4,7-*S*,*S*-bi-GS-DHI having an absorption peak at 280 nm.

The SN is considered to have a high rate of metabolism because it contains high levels of oxidizable substances such as DA, highly unsaturated fatty acids, Fe(III) ions and relatively fewer amounts of antioxidants such as GSH [39,40]. Therefore, it is considered that ROS are easily generated under such oxidative stress and results in cell death in the SN. Although various catecholamines seem to be involved in the synthesis of NM, either DA or NE may be involved in the oxidative polymerization in the SN or most other brain areas, including the LC, respectively [4,41]. Thus, the existence of Fe(III) is important in the biosynthesis of NM, which strongly promotes the auto-oxidation of catecholamines into reactive quinones [20]. Sun et al. demonstrated that the formation of DA bound Fe(III) and Fe(II) complexes as well as the cyclization and rearrangement of DA-derived quinones are the most important pathways of DA metabolism, with each process heavily dependent on pH [42]. Their experiments also indicated that, in the presence of the same concentrations of DA and H_2_O_2_, acidosis in PD brain results in an increase in DAC accumulation and DA-mediated productions of hydroxyl radicals, with these potential toxicants likely to further aggravate the progression of PD and severity of symptoms arising from DA denervation. Using tyrosinase, Fe(III) or Cu(II) as oxidative reagents, we prepared water-soluble protein conjugates from DPRA(Cys) as a peptide model, BSA as a representative protein model and βLG as a protein model to be converted into an amyloid fibrillar form (fLG) [29]. By comparing the oxidation reactions of DA with DPRA(Cys) and DPRA(Lys), we confirmed that DAQ easily binds via SH groups compared with NH_2_ groups. The oxidation of DA by tyrosinase in the presence of BSA proceeded rapidly and Cys-DA isomers were obtained in a combined yield of 46% after HCl hydrolysis of the reaction product.

The present study shows the relative reactivity of protein thiols and amines (imidazole and -NH_2_) against DAQ in the following order: BSA-SH > the side chain amino group of DAQ (cyclization) > βLG-NH_2_ (and possibly BSA-NH_2_) > βLG-SH. BSA-SH is ~10 times more reactive than βLG-SH, based on the yields of Cys-MeCA (66% for BSA and 16% for βLG) and SH contents (0.30 for BSA and 0.59 for βLG). The reason for the order βLG-NH_2_ > βLG-SH is that while βLG appears to react quantitatively with MeBQ (Figure 2F), the yield of Cys-MeCA that originates from βLG-SH is only 16%. The present study also shows that DAQ reacts with protein thiols to produce catechol-protein conjugates in varying yields (Figure 8). The binding of DA and proteins go through different pathways depending on the structural differences of the protein (Figure 8). In the oxidative reaction between DA and BSA, the differential spectral method revealed that 70% of DAQ proceeded via Pathway 1, 46% of which proceeded via protein-bound DA through the Cys residue, which was confirmed by HPLC analysis. The remaining 24% of DAQ became DHI-protein via DAC protein. The reduced forms of the catechol-protein conjugates remain relatively stable (46% in BSA and 2.1% in βLG), as shown in Figure 5B. Some portions of the catechol-protein conjugates are oxidized through redox reactions to form quinone-protein conjugates, which are cyclized to give the DAC-protein-1 and are then oxidized to melanochrome-protein conjugate 1 (24% in BSA and 18% in βLG). On the other hand, the remaining 30% of DAQ produced DAC, about 21% of which progresses via DAC-protein-2 in Pathway 2, finally giving melanochrome-protein conjugate 2 (absorbance peaks of 370 and 570 nm). The remaining ~9% of DAC progresses via Pathway 3 to produce DHI-melanin. From βLG, the possible fate of DAQ is more speculative. Only 20% of the DAQ appears to react with βLG to form the catechol-protein. βLG bound via its Cys residue with DA at a 2% yield, and the remaining 18% might become melanin-protein conjugates probably via NH_2_ residues of proteins. The remaining 80% of DAQ is converted to DAC, the majority of which may be rearranged to DHI, finally giving DHI-melanin via Pathway 3. DHI-melanin was obtained in a yield of ~64%. The remaining minor portion (about 16%) of DAC reacts with βLG to form melanochrome-protein conjugate 2 (absorbance peaks of 370 and 590 nm).

The thiol group in Cys residues of proteins is much more reactive compared to the imidazole group in His residues and the terminal amino group in Lys residues of proteins. This high reactivity of the thiol group in proteins has been reported by Li et al. [30]. The Cys residue in BSA is reactive enough to conjugate with DAQ. Even if BSA escapes from the conjugation with DAQ, it reacts with DAC through the Cys residue. On the other hand, the Cys residue in βLG is not so reactive as the Cys residue in BSA. βLG reacted with MeBQ quantitatively, as judged by the absorption spectra (Figure 2F). Nevertheless, the yield of Cys-MeCA in the hydrolysate was only 16% (Figure 5A). These results indicate that a majority (ca. 80%) of the conjugation with MeBQ takes place with the His and Lys residues in βLG. It has not been fully clarified whether the conjugation with DAC proceeds through protein-SH or protein-NH_2_. However, the thiol group of BSA (and possibly βLG) is likely to participate in the conjugation, because the amino groups in His and Lys do not react with DAC (data not shown). Li et al. reported that at pH 7.0, the α-carbon amino group of Lys reacted faster with MeBQ than the side chain amino group because of lower pKa value of the former [30]. We observed the same reactivity of Lys (2 eq.) and NAcLys (10 eq.) with MeBQ. This observation showed that Lys with free α-carbon amino group reacted faster than NAcLys with α-carbon amino group protected by acetyl group, and this was the same result with Li et al.

The exact structures of the melanochrome-protein conjugates are unknown. The quinone form of DHI, 5,6-indolequinone, is extremely reactive and dimerizes with DHI to form DHI dimers [43,44,45]. However, if highly reactive nucleophiles such as thiols and melanin intermediates such as DHI are absent in the reaction medium, those melanochrome-protein conjugates may survive for hours.

The absorption spectra in the oxidation of DA+BSA and DA+ βLG by tyrosinase showed peaks derived from a DAC structure with an absorption maximum around 470 nm in 1 min (Figure 4), and catechol-protein from the oxidation of DAQ and protein gradually changed to protein-bound melanin having a DHI moiety with the stable *o*-quinone structure showing absorption peaks at 370 nm and 570-590 nm. Compared to the tyrosinase reaction with DA and BSA as well as with βLG, it was shown that reactions with metal ions having redox activity proceeded gradually although the yields and reaction rates were different. Therefore, the results suggest that the subsequent reaction pathway depends on the structural characteristics of the protein toward the affinity of metal ions. This study confirmed that DA oxidatively binds via Cys residues of proteins or peptides. This difference could be due to the reduced reactivity of βLG with DAQ by steric hindrance and electronic interactions as well as differences in the pKa of the thiols, as thiolate anions are better nucleophiles than neutral species [46].

In contrast to the tyrosinase-catalyzed oxidation, the redox-active metal-catalyzed oxidation gives quantitative yields of BSA-*S*-DA conjugates formed through the Cys residue when we take the remaining DA into account (Figure 5). This could be due to the much slower oxidation of DA to DAQ by Fe(III) or Cu(II) compared to the oxidation by tyrosinase. The DA-protein conjugates remain in the reduced, catecholic forms, which are rather stable. This high efficacy of binding is important when we consider the in vivo situation in which Fe(III) most likely plays a major role of oxidizing DA [3,47]. Fe(III) and Cu (II) in the absence of EDTA do not function as true catalysts in the oxidation reaction of catecholamines, but are consumed in the process, integrated into NM. In the presence of EDTA the cations are sequestered, but still redox active in the reaction [47].

This study demonstrates that protein-DA conjugates have pro-oxidant activity without metal ions and generate ROS, suggesting that they are involved in neurodegenerative diseases such as PD. The reduced state of melanin is able to reduce oxygen molecules and the superoxide radicals generated react with GSH, and thereafter a radical reaction proceeds. Superoxide radicals dismutate to molecular oxygen and H_2_O_2_, and the thiol group of GSH is oxidized to GSSG by superoxide radicals. Based on this oxidative radical reaction, we measured the pro-oxidant activity by measuring GSH/GSSG and H_2_O_2_ production [48]. As a control, we measured the pro-oxidant activity of DA-melanin and oxidative products from BSA and βLG alone, but those compounds did not show any pro-oxidant activities. Our study provides evidence that protein-DA conjugates possess pro-oxidant activities, and the oxidation products (quinones, aminochromes and DHI-derivatives) from GSH-DA, BSA-*S*-DA and βLG-*S*-DA were shown to exhibit several times more potent redox activities than Cys-DA melanin (Figure 7). It is known that pheomelanins such as cysteinyldopa-melanin exhibit potent pro-oxidant activities [49]. Oxidation products from BSA and βLG alone produced small amounts of GSSG. This could be ascribed to the production of DOPA residues when proteins are oxidized by tyrosinase [50].

## 4. Materials and Methods

### 4.1. Materials

Tyrosinase (from mushrooms, specific activity 1715 U/mg), Ampliflu™ Red reagent (1-acetyl-3,7-dihydroxyphenoxazine), DA, BSA, βLG, MeCA, l-Cys and NAC were purchased from Sigma-Aldrich (St. Louis, MO, USA). FeCl_3_, phenol, 3,5-di-*tert*-butyl-1,2-benzoquinone (DBBQ), 5,5′-dithiobis(2-nitrobenzoic acid) (DTNB), 2-mercaptoethanol (thioglycol), reduced glutathione (GSH), thioglycolic acid, n-hexane, H_2_O_2_, EDTA·2Na and methanol (HPLC grade) were purchased from Wako Pure Chemical Industry (Osaka, Japan). DPRA(Cys) (Ac-RFAACAA-COOH) and DPRA(Lys) (Ac-RFAAKAA-COOH) were purchased from Scrum Inc. (Tokyo, Japan). CuSO_4_·5H_2_O and HClO_4_ were from Katayama Chemical Co. Ltd. (Osaka, Japan). NAcHis, NAcLeu, NAcLys and dimethyl sulfoxide (DMSO) were from Tokyo Chemical Industry Co. (Tokyo, Japan). All other chemicals were of the highest purity commercially available. The highest purity Milli-Q water (Milli-Q Advantage, Merck Millipore Co., Tokyo, Japan) was used in order to avoid contamination of metal ions. The number of thiol group in BSA and βLG were determined to be 0.30 and 0.59 mol/mol protein, respectively, using the method of Elleman [51] with DTNB. The number of thiol group in BSA was close to 0.38 mol/mol BSA [30].

### 4.2. Instruments

For measurement of H_2_O_2_, the maximum absorption wavelength of 568 nm exhibited by oxidized red fluorescent resorufin in Ampliflu™ Red was used.

Ultraviolet-visible (UV-Vis) spectra were measured using a JASCO V-630 UV-VIS spectrophotometer (JASCO Co., Tokyo, Japan).

The HPLC system used consisted of an analytical UV-Vis detector (JASCO Co.), a JASCO pump (JASCO Co.) and a Shiseido C18 column (Capcell Pak MG; 4.6 mm × 250 mm; 5 µm particle size, Shiseido, Tokyo, Japan). For the measurement of Cys-DA isomers and Cys-MeCA isomers, the mobile phases were 0.4 M HCOOH:MeOH = 99:1 and 80:20, respectively. The flow rate was 0.7 mL/min. The column temperature was 35 °C for Cys-DA isomers and 45 °C for Cys-MeCA isomers.

^1^H-nuclear magnetic resonance (NMR (400 MHz) spectra were obtained using a Bruker AVANCE 400 spectrometer (Billerica, MA, USA) and are reported as the chemical shift d (ppm) downfield from sodium 2,2-dimethyl-2-silapentane-5-sulfonate used as an internal reference. Coupling constants (J) are expressed in Hz and signals are expressed as s (singlet), d (doublet) or t (triplet).

### 4.3. Tyrosinase-Catalyzed Oxidation of DA in the Presence of BSA or βLG and HPLC Estimates of Binding through the Cys Residue

Tyrosinase (100 U/20 μL) was added to a stirring solution of 20 μL 10 mM DA (final concentration 0.1 mM) and 400 μL 1 mM BSA or 1 mM βLG (0.2 mM) in 1560 μL 0.05 M sodium phosphate buffer, pH 7.4, at 37 °C in 10 ml screw-capped conical glass tubes. After predetermined times (0, 3 and 60 min), 100 μL of each reactant was added to a 10 ml screw capped conical glass test tube containing 10 μL 10% NaBH_4_ followed by the addition of 10 μL 6 M HCl and 20 μL buffer. After vortex mixing, 500 μL 6 M HCl containing 5% thioglycolic acid and 1% phenol was added and the mixture was hydrolyzed for 20 h at 110 °C under an argon atmosphere. After cooling, 130 μL of each reaction mixture were transferred to 1.5-mL micro-tubes containing 50 mg alumina and 200 μL 1% Na_2_S_2_O_5_-1% EDTA·2Na. To this, 700 μL 2.7 M Tris·HCl (pH 9.0)-2% EDTA·2Na was added and shaken vigorously for 5 min, followed by centrifugation for 15 s at 10,000 rpm. The upper layer was removed using an aspirator and the alumina was washed with 1 mL Milli-Q water followed by aspiration of the upper layer after centrifugation for 15 s at 10,000 rpm. This procedure was performed three times. After shaking the alumina vigorously in 200 μL 0.4 M HClO_4_ for 2 min followed by centrifugation for 15 s at 10,000 rpm, the contents of Cys-DA isomers were measured by HPLC analysis of the supernatants. For the standard Cys-DA isomers, 100 μL 100 μM standard solution containing Cys-DA isomers (5-*S*-Cys-DA, 2-*S*-Cys-DA, 6-*S*-Cys-DA and 2,5-*S*,*S’*-diCys-DA) were added to a 10 ml screw capped conical glass test tube containing 10 μL 10% NaBH_4_ followed by the addition of 10 μL 6 M HCl and 20 μL (20 nmol) BSA or βLG. After vortex mixing, 500 μL 6M HCl containing 5% thioglycolic acid and 1% phenol were added and the mixture was hydrolyzed for 20 h at 110 °C under an argon atmosphere. After cooling, the following operations were performed in the same way as described above. The yields were obtained by comparing peak heights to the standard. Retention times of 2-*S*-Cys-DA, 6-*S*-Cys-DA, 5-*S*-Cys-DA and 2,5-*S*,*S’*-diCys-DA are 5.5, 5.8, 9.9 and 19 min, respectively, in a typical HPLC chromatogram of total Cys-DA isomers from hydrolysates of DA–BSA conjugate.

The tyrosinase-catalyzed oxidation of MeCA (final concentration 0.1 mM) in the presence of BSA or βLG (0.2 mM) was similarly performed and analyzed for Cys-MeCA isomers. For Cys-MeCA isomers, the peak height of 4-*S*-Cys-5-MeCA was 1.47-fold higher than that of 3-*S*-Cys-5-MeCA and this difference was compensated in calculation of the yields. Retention times of 4-*S*-Cys-5-MeCA, 3-*S*-Cys-5-MeCA, and diCys-MeCA are 8.0, 12.0, and 14.3 min, respectively.

### 4.4. Fe (III) or Cu (II)-Catalyzed Oxidation of DA in the Presence of BSA or βLG and HPLC Estimates of Binding through the Cys Residue

Twenty μL 10 mM FeCl_3_ (or 10 mM CuSO_4_·5H_2_O) (final concentration 0.1 mM) was added to a shaking solution of 20 μL 10 mM DA (0.1 mM) and 400 μL 1 mM BSA (0.2 mM) in 1540 μL 0.05 M sodium phosphate buffer, pH 7.4, containing 20 μL 10 mM EDTA·2Na at 37 °C in a screw-capped glass tube. After predetermined times (0, 1, 2, 3 and 4 h), 100 μL of each reaction mixture were subjected to HCl hydrolysis followed by HPLC analysis of Cys-DA isomers.

### 4.5. Pro-Oxidant Activity of Various Melanins Derived Oxidatively from DA-Thiol Conjugates

A mixture of 100 μL 10 mM DA (final concentration 1 mM) and 100 μL 10 mM Cys (or 10 mM GSH) (1 mM) were oxidized using 20 μL tyrosinase (5,000 U/mL) in 780 μL 0.05 M sodium phosphate buffer, pH 7.4, at 37 °C for 4 h. After oxidation, 50 μL 10 mM GSH were added to 500 μL oxidation mixture, and the reaction was continued for 30 and 60 min. For BSA and βLG, 50 μL 10 mM DA and 500 μL 1 mM BSA (or 1 mM βLG) were oxidized using 10 μL tyrosinase (5,000 U/mL) in 440 μL 0.05 M sodium phosphate buffer, pH 7.4, at 37 °C. After oxidation for 4 h, 25 μL 10 mM GSH were added to 500 μL oxidation mixture, and the reaction was continued for 30 and 60 min. Subsequently, all of the reaction mixture was separated by centrifugation at 10,000 rpm for 1 min and the supernatant was used for measurement.

For the determination of H_2_O_2_, H_2_O_2_ was analyzed spectrophotometrically after reaction with the chromogen Ampliflu™ Red to form a red pigment having an absorption maximum at 568 nm closely following the manual (Invitrogen) [52]. Briefly, the chromogen solution was prepared by adding Ampliflu™ Red solution (1.54 mg in 0.6 mL DMSO, 50 µL) and horseradish peroxidase (100 U/mL, 100 µL) to 50 mM sodium phosphate buffer, pH 7.4 (4.85 mL). Before analysis, a 50 µL aliquot of each sample solution was diluted with 150 µL 0.05 M sodium phosphate buffer, pH 7.4, (as a standard solution, H_2_O_2_ was diluted with buffer solution to a final concentration of 10 µM). A 200 µL aliquot of the mixture was mixed with the chromogen solution (200 µL) and the mixture was left at room temperature for 10 min. Absorption spectra were measured between 450 to 650 nm. Backgrounds for the oxidized reagent and melanin were subtracted using absorption spectra.

For the determination of GSH and GSSG, GSH was analyzed using our HPLC method [37,49,53]. We used 50 µL of each oxidation mixture followed by dilution with 950 µL 0.4 M HClO_4_. One hundred µL of each mixture was mixed with a freshly prepared ethanol solution of 100 µL 1 mM DBBQ and was shaken for 30 min at 30 °C. The mixture was then extracted with hexane (1 mL) by shaking for 2 min to remove unreacted DBBQ. This process was repeated three times. BSA mixtures were further centrifuged at 10,000 rpm for 1 min using a filtration tube (0.45 µm, PVDF/G, Alltech^®^, Alltech Associates. Inc., Deerfield, IL, USA) in order to remove the turbidity of the mixture. The resultant solution (20 µL) was injected directly into the HPLC system after centrifugation. The HPLC system was modified from the original conditions as follows. A mobile phase of 0.4 M HCOOH: methanol, 30:70 (*v*/*v*) was used with a UV detector at 294 nm and a column temperature of 45 °C. GS-DBBQ and Cys-DBBQ adducts eluted at 10.4 and 12.0 min, respectively.

For GSSG measurement, 20 µL 5 mM 2-mercaptoethanol were added to 20 µL of the oxidation mixture and were mixed. Twenty µL 0.4 M Na_2_CO_3_ were then added and were shaken for 30 min at 30 °C. Next, 40 µL 1.2 M HClO_4_ were added and reacted with ethanol solution of 100 µL 1 mM DBBQ in the same manner as GSH. A standard solution containing 50 µM each of GSH and Cys (100 µL) or 25 µM GSSG and Cystine (20 µL) were similarly treated.

### 4.6. Preparation of 4-S-Cysteinyl-5-Methylcatechol

3-*S*-Cysteinyl-5-methylcatechol was prepared through a reaction between 4-methyl-*o*-quinone and l-Cys [54]. 4-*S*-cysteinyl-5-methylcatechol was a hitherto non-described compound and was prepared through a reaction of MeCA with l-cystine in aqueous HBr [55] as follows. A solution of 2.48 g (20 mmol) MeCA and 1.20 g (5 mmol) l-cystine in 50 mL 47% HBr was heated under reflux for 2 h. After cooling, the mixture was evaporated to dryness under reduced pressure. The residue was dissolved in 10 mL 1 M HCl and was extracted 3 times with 20 mL ethyl acetate. The aqueous layer was evaporated to dryness and the residue was applied on a column of Dowex 50W × 2 (1.8 cm × 15 cm, equilibrated in 2 M HCl). The column was eluted with 2 M HCl, and fractions of 10 mL were collected and monitored by UV spectrophotometry. Evaporation of fractions 21–26 and 31–40 afforded 286 and 179 mg of the HCl salts of 4-*S*-cysteinyl-5-methylcatechol and 3-*S*-cysteinyl-5-methylcatechol, respectively. Their solutions in 1% sodium metabisulfite were neutralized (pH ca. 6) with sodium acetate (solid) to afford 68.6 mg (4.9% from l-cystine) and 28.2 mg (2.0%) of the free amino acids as colorless crystals. We did not try to improve the recovery of the free amino acids. UV in 0.1 M HCl of 4-*S*-cysteinyl-5-methylcatechol: λmax 251 nm (ε 4800) and 292 nm (ε 3200).

^1^H-NMR (2M DCl) of 4-*S*-cysteinyl-5-methylcatechol: δ2.33 (3H, s), 3.39 (1H, dd, J = 14.8, 6.0), 3.44 81H, dd, J = 14.8, 6.0), 4.28 (1H, t, J = 6.0), 6.85 (s, 1H), 7.10 (1H, s). ^1^H-NMR (2M DCl) of 3-*S*-cysteinyl-5-methylcatechol: δ2.40 (3H, s), 3.33 (1H, dd, J = 15.2, 4.8), 3.46 (1H, dd, J = 15.2, 4.8), 4.24 (1H, t, J = 4.8), 6.80 (1H, d, J = 8.0), 6.92 (1H, d, J = 8.0). HPLC: 0.4 M HCOOH:MeOH = 80:20, retention time: 4-*S*-cysteinyl-5-methylcatechol: 8.0 min, 3-*S*-cysteinyl-5-methylcatechol: 12.0 min.

## 5. Conclusions

We prepared a variety of thiol-bound DA complexes and showed that DA binds via the Cys residue of proteins and that the efficacy of binding depends on the structural features of the proteins. Protein-conjugated DA-derived melanin can oxidize GSH to GSSG with the concomitant production of H_2_O_2_. These results suggest that oxidatively modified DA-protein conjugates produced in the brain possesses a potent pro-oxidant activity, which may cause neurodegeneration through the production of ROS and the depletion of antioxidants.

## Figures and Tables

**Figure 1 ijms-20-02575-f001:**
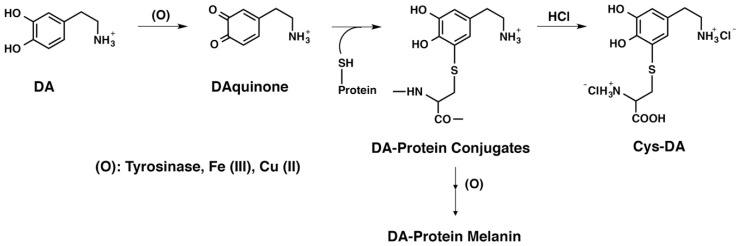
HCl hydrolysis of l-dopamine (DA)-protein conjugates and DA-protein melanin. Binding via Cys residues in proteins was formed by oxidation using Fe(III) or Cu(II) as well as tyrosinase.

**Figure 2 ijms-20-02575-f002:**
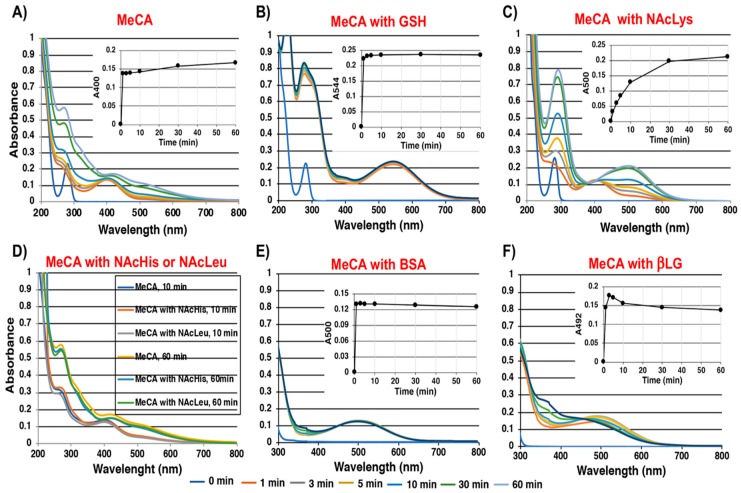
Time course of tyrosinase-catalyzed oxidation of (**A**) 4-methylcatechol (MeCA), (**B**) MeCA with glutathione (GSH), (**C**) MeCA with 10 eq. *N*α-acetyl-l-lysine (NAcLys), (**D**) MeCA with 10 eq. *N*-acetyl-l-histidine (NAcHis) or 10 eq. *N*-acetyl-l-leucine (NAcLeu), (**E**) MeCA with bovine serum albumin (BSA), and (**F**) MeCA with β-lactoglobulin (βLG). The inserted figures show the kinetic monitoring at indicated absorption wavelength (nm).

**Figure 3 ijms-20-02575-f003:**
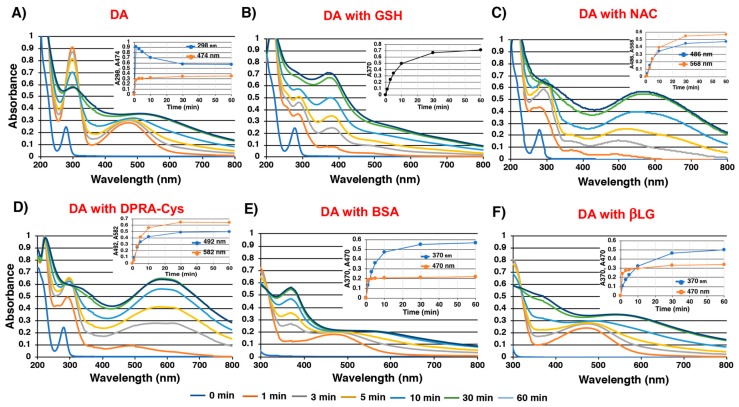
Time course of tyrosinase-catalyzed oxidation of (**A**) DA, (**B**) DA with GSH, (**C**) DA with *N*-acetyl-l-cysteine (NAC), (**D**) DA with DPRA(Cys) (a heptapeptide containing 1 Cys residue, Ac–RFAACAA–COOH), (**E**) DA with BSA, and (**F**) DA with βLG. The inserted figures show the kinetic monitoring at indicated absorption wavelength (nm).

**Figure 4 ijms-20-02575-f004:**
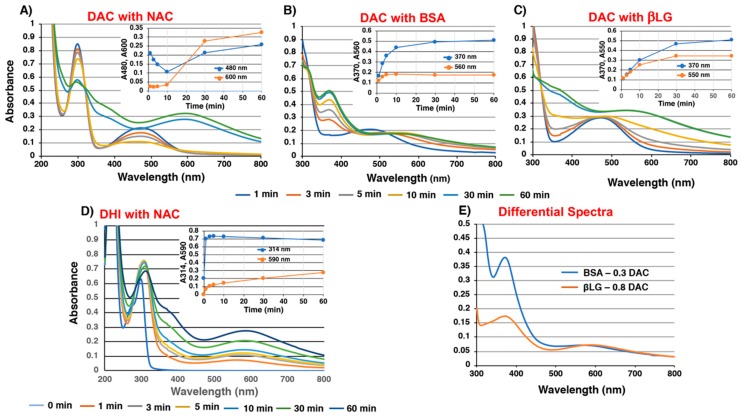
Time course of tyrosinase-catalyzed reaction of (**A**) DAchrome (DAC) with NAC, (**B**) DAC with BSA, and (**C**) DAC with βLG, and tyrosinase-catalyzed oxidation of (**D**) 5,6-dihydroxyindole (DHI) with NAC. (**E**) Differential spectra for BSA 70% and βLG 20% (see the text). The inserted figures show the kinetic monitoring at indicated absorption wavelength (nm).

**Figure 5 ijms-20-02575-f005:**
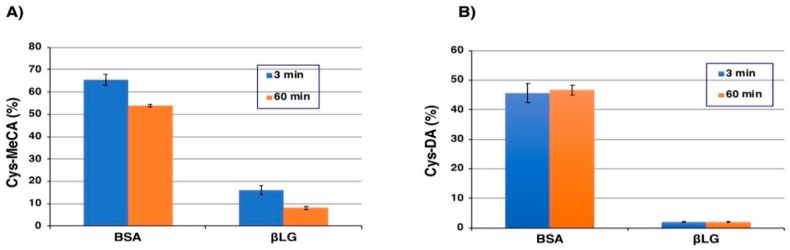
(**A**) Combined yield of Cys-MeCA isomers (3-*S*-Cys-5-MeCA, 4-*S*-Cys-5-MeCA and diCys-MeCA) after HCl hydrolysis of MeCA conjugates with BSA or βLG after 3 min and 60 min. (**B**) Combined yield of Cys-DA isomers (5-*S*-Cys-DA, 2-*S*-Cys-DA, 6-*S*-Cys-DA and 2,5-*S*,*S’*-diCys-DA) after HCl hydrolysis of DA conjugates with BSA or βLG after 3 min and 60 min. Data are means (*n* = 4) and standard error of the mean (SEM) from 2 experiments, each hydrolysate being analyzed in duplicate.

**Figure 6 ijms-20-02575-f006:**
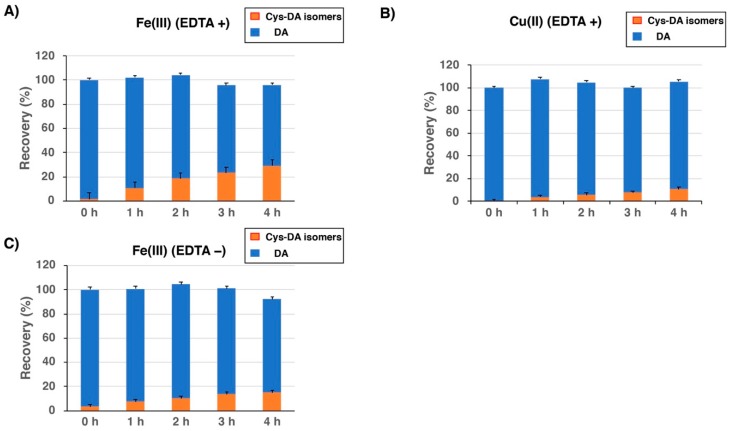
Time course of HCl hydrolysis of conjugates obtained from oxidation of DA and BSA with (**A**) Fe(III) in the presence of EDTA, (**B**) Cu(II) in the presence of EDTA, and (**C**) Fe(III) in the absence of EDTA. Yields of Cys-DA isomers are those from 5-*S*-Cys-DA, 2-*S*-Cys-DA, 6-*S*-Cys-DA and 2,5-*S*,*S’*-diCys-DA combined. Data are means (*n* = 4) and SEM from 2 experiments, each hydrolysate being analyzed in duplicate.

**Figure 7 ijms-20-02575-f007:**
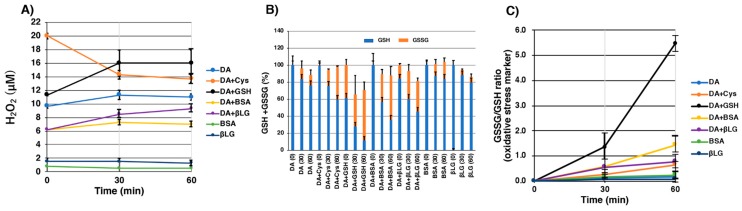
Measurement of pro-oxidant activity of DA-thiol conjugates (melanins) produced by tyrosinase oxidation at pH 7.4 for 4 h. Cys, GSH, BSA and βLG were used as thiols. (**A**) H_2_O_2_ production during incubation with GSH for 30 and 60 min. (**B**) Changes in GSH and oxidized glutathione (GSSG) contents during incubation with GSH for 30 and 60 min. (**C**) The ratio of GSSG to GSH (an oxidative stress marker) during incubation with GSH for 30 and 60 min. Data are means and SEM from 3 experiments.

**Figure 8 ijms-20-02575-f008:**
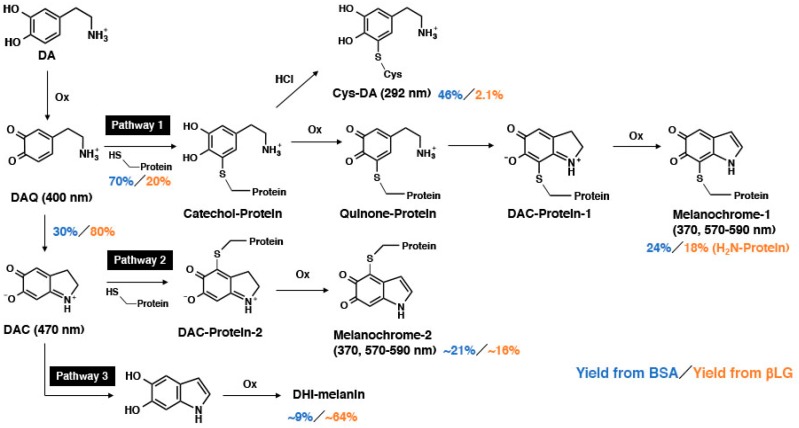
Possible reactions of dopamine quinone (DAQ) and DAC with proteins, BSA (2 eq. containing 0.6 eq. of thiol) and βLG (2 eq. containing 1.2 eq. of thiol). DAQ reacts with thiol proteins to form Catechol-protein (Pathway 1), which are oxidized to Melanochrome-1. Some portions of DAQ escape from reaction with proteins and are converted to DAC (Pathway 2). DAC is able to react with thiol proteins to form melanochrome-2. Melanochrome-1 and -2 are isomers differing in the position of conjugation with proteins. Some portions of DAC escape from reaction with proteins and are converted to DHI-melanin (Pathway 3).

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
