# Peer review of "The Oxidative Pathway to Dopamine–Protein Conjugates and Their Pro-Oxidant Activities: Implications for the Neurodegeneration of Parkinson’s Disease"

_ijms, 2019, doi:10.3390/ijms20102575_

Reviewer 1 Report

This is an interesting study of the neurochemistry of dopamine oxidation, attempting to shed some light on the susceptibility of dopaminergic neurones in the pathogenesis of Parksinson's disease. The authors used an in vitro strategy of measuring the DA-protein conjugates in the presence of target proteins that contain cysteine, the amino acid most likely to surrender protons to DA quinone oxidation to produce compounds that are detectable by various methods including absorption spectrophotometry and HPLC. They also measured the ability of various synthetic melanins to generate hydrogen peroxide in solution as a marker of pro-oxidant activity.

The study is presented in a clear fashion, and contributes to the field. It would be good to know which preparation method was used in the BSA, as the exposure of the Cys residues is determined by the oxidation state of the Cys residues. Some discussion about the selection of mushroom tyrosinase would add to the study. I understand the rationale for the use of tyrosinase in these studies, i.e. to catalyse the reactions, but human tyrosinase is much more limited in its ability to use catechols such as dopamine as a substrate to produce melanin.  The rationale for EDTA should be clearer. My understanding is that Cu2+ and Fe3+ in the absence of EDTA do not function as true catalysts in the oxidation reaction of catecholamines, but are consumed in the process, integrated into NM.  The presence of EDTA ensures that the cations are sequestered, but still redox active in the reaction. Perhaps this could be discussed. 

Did the authors consider other methods to detect DAquinone protein conjugates such at colorimetric or changes in fluorescence?

Overall a nice study that contributes to the field.

Author Response

We really appreciate the comments and suggestions from the Associate Editor and the Reviewers. We found them to be very helpful and constructive and we modified the manuscript accordingly. The revised text is highlighted in yellowin the attached manuscript.

Reviewer(s)' Comments to Author:

Reviewer: 1

Comments to the Author:

1. Some discussion about the selection of mushroom tyrosinase would add to the study. I understand the rationale for the use of tyrosinase in these studies, i.e. to catalyse the reactions, but human tyrosinase is much more limited in its ability to use catechols such as dopamine as a substrate to produce melanin.

              Response: We really appreciate the favorable comments by Reviewer 1. In this study we just used the mushroom tyrosinase for the purpose of oxidizing DA to DAquinone. As the reviewer mentioned, I think that the involvement of human tyrosinase is controversial, but we will not touch on human tyrosinase in this paper.

2.The rationale for EDTA should be clearer. My understanding is that Cu2+and Fe3+in the absence of EDTA do not function as true catalysts in the oxidation reaction of catecholamines, but are consumed in the process, integrated into NM. The presence of EDTA ensures that the cations are sequestered, but still redox active in the reaction. Perhaps this could be discussed.

              Response: We really appreciate the favorable comments by Reviewer 1. According to the Reviewer’s suggestion, we added the sentence ‘Fe(III) and Cu (II) in the absence of EDTA do not function as true catalysts in the oxidation reaction of catecholamines, but are consumed in the process, integrated into NM. In the presence of EDTA the cations are sequestered, but still redox active in the reaction [48].

48. Zecca, L.; Casella, L.; Albertini, A.; Bellei, C.; Zucca, F. A.; Engelen, M.; Zadlo, A.; Szewczyk, G.; Zareba, M.; Sarna, T. Neuromelanin can protect against iron-mediated oxidative damage in system modeling iron overload of brain aging and Parkinson’s disease. J. Neurochem. 2008, 106, 1866-1875.

3. Did the authors consider other methods to detect DAquinone protein conjugates such as colorimetric or changes in fluorescence?

Response: We did not use other methods such as colorimetric or changes in fluorescence to detect DAquinone, but UV-visible spectra.

Reviewer 2 Report

The manuscript by Wakamatsu et al investigates the pathways by which dopamine can modify protein residues. The authors study the effect of DA quinone- mediated amino acid modification under various conditions and provide novel data on the chemistry of these modifications as well as their ability to be the source of oxidative stress. The paper is well written and, except for one major point (see below), conclusions are supported by the data. There are also several changes that the authors can make to the figures that, in my opinion, will help the readers.

Major:

1. Line 217. “We then examined whether H2O2 is produced during the oxidation of GSH by those synthetic melanins.” Why is it expected that H2O2 will be formed during GSH oxidation? The opposite is in fact expected and that brings the question of why there is no correlation between GSH/GSSG and H2O2 concentration on Fig 7? Was peroxidase added to the reaction mixture when Ampliflu Red assay was performed?

2. Line 215. “GSH (1 mM) was exposed to DA-thiol conjugates prepared by tyrosinase oxidation of DA (1 mM) in the presence of Cys (1 mM) or GSH (1 mM), after which the amounts of remaining GSH and oxidized glutathione (GSSG) were analyzed by our specific HPLC method after 30 min or 60 min [37].” Was the amount of GSH remaining after DA oxidation reaction measured and accounted for? This could significantly affect the results shown on Fig 7A.

Minor:

1. Instead of (or in addition to) showing the raw spectroscopy data on Figs 2-4, show time kinetics of changes of absorbances at specific wavelengths. Also, it will be helpful if chemical structures of compounds that absorb at indicated wavelengths are shown.

2. Better to show data on Fig 7 as time kinetics (curves) rather than bar graphs. Also, GSH to GSSG ratios will better demonstrate the pro-oxidative potential of different DA derivatives.

Author Response

We really appreciate the comments and suggestions from the Associate Editor and the Reviewers. We found them to be very helpful and constructive and we modified the manuscript accordingly. The revised text is highlighted in yellow in the attached manuscript.

Reviewer(s)' Comments to Author:

Reviewer: 2

Comments to the Author:

1. Line 217. “We then examined whether H2O2is produced during the oxidation of GSH by those synthetic melanins.” Why is it expected that H2O2will be formed during GSH oxidation? The opposite is in fact expected and that brings the question of why there is no correlation between GSH/GSSG and H2O2concentration on Fig 7? Was peroxidase added to the reaction mixture when Ampliflu Red assay was performed?

              Response: We really appreciate the favorable comments by Reviewer 2. Melanin pigment undergoes redox-cycling, giving rise to superoxide anion production and oxidation of reducing agents such as GSH. GS·radicals immediately dimerize to give oxidized glutathione (GSSG) (Ito, S. et al., Photochem. Photobiol.,94, 409-420, 2018). We also have already published the paper regarding the pro-oxidant activity that the quinone is very reactive and its oligomer can oxidize GSH to GSSG with a concomitant production of hydrogen peroxide, indicating its pro-oxidant activity (Ito, S. et al., Chemical Research Toxicology, 30, 859-868, 2017). Figure 7A shows the amount of H2O2after the addition of GSH to melanin prepared in situ. Compared with Figure 7A and 7C, we could observe that the amount of H2O2and the ratio of GSSG to GSH gradually increased with time. Regarding Ampliflu Red assay, we used horseradish peroxidase shown in Materials and Methods. It is known that the Ampliflu Red Hydrogen peroxide assay kit contains a sensitive, one-step assay that uses the Ampliflu Red reagent to detect H2O2or peroxidase activity, and in combination with horseradish peroxidase, has been used to detect H2O2released from biological samples, including cells or generated in enzyme-coupled reactions.

2. Line 215. “GSH (1 mM) was exposed to DA-thiol conjugates prepared by tyrosinase oxidation of DA (1 mM) in the presence of Cys (1 mM) or GSH (1 mM), after which the amounts of remaining GSH and oxidized glutathione (GSSG) were analyzed by our specific HPLC method after 30 min or 60 min [37].” Was the amount of GSH remaining after DA oxidation reaction measured and accounted for ? This could significantly affect the results shown on Fig 7A.

              Response: Yes, we measured the GSH remaining and GSSG produced. GSH was added to DA-thiol conjugates prepared by tyrosinase oxidation of DA in the presence of Cys, GSH, BSA or βLG. We measured GSH and GSSG (0 min) as soon as GSH is added to the oxidation mixtures, and analyzed the amount of GSH remaining and GSSG produced at 30 min and 60 min compared with the original amount of GSH at 0 min.

3. Instead of (or in addition to) showing the raw spectroscopy data on Figs 2-4, show time kinetics of changes of absorbances at specific wavelengths. Also, it will be helpful if chemical structures of compounds that absorb at indicated wavelengths are shown.

              Response: We really appreciate the favorable comments. According to the reviewer’s suggestion, we drew new figures on Figure 2 – 4 containing the inserted figures showing the kinetic monitoring at indicated absorption wavelength (nm). We did not show the chemical structures of compounds that absorb at indicated wavelength because we cannot assign the accurate chemical structures although we can say the structures as melanochrome-type compound or ortho-quinones.

4. Better to show data on Fig 7 as time kinetics (curves) rather than bar graphs. Also, GSH to GSSG ratios will better demonstrate the pro-oxidative potential of different DA derivatives.

              Response: We really appreciate the favorable comments. According to the reviewer’s suggestion, we drew new figures on Figure 7A and 7C. We prepared the Figure 7A of time kinetics showing H2O2production during incubation with GSH for 30 min and 60 min. Figure 7C shows the ratio of GSSG to GSH as an oxidative stress marker during incubation with GSH for 30 min and 60 min.